



# Investigating the Dust Flux in the Meteoric Smoke Sampler (MESS) Instrument for Sampling Dust in the Mesosphere

Henriette Trollvik[1], Ingrid Mann[1], Sveinung Olsen[1], and Yngve Eilertsen[1]

[1]UiT - The Arctic University of Norway, 9019 Tromsø

**Correspondence:** Henriette Trollvik (henriete.m.trollvik@uit.no)

**Abstract.** We report and discuss the design of a rocket instrument to collect mesospheric dust particles that are composed of ice and include smaller refractory meteoric smoke particles (MSP). We expect that the ice components melt and that MSP are collected. The instrument consists of a collection device with an opening and closure mechanism and an attached conic funnel. Attaching the funnel increases the sampling area in comparison to the collection area which is kept small since this determines

the size of the closure device which is a critical component to be designed for sea recovery. The instrument will collect primary particles that directly hit the collection area and secondary particles that form from mesospheric dust hitting the funnel. We simulate the entry and impact of dust onto the detector considering their trajectories in the airflow and the fragmentation at the funnel. We estimate the collection efficiency of the instrument and the impact energy of particles at the collecting area. The design considered has a sampling area of 5 cm diameter and a collection area of 1.8 cm diameter. To estimate the expected

amount of collected dust we assume collection during rocket flight through a 0.5 to 4 km dust layer with dust number densities and dust sizes at 85 km as derived from lidar observations (Kiliani et al. 2015). Assuming the collected particles contain 3% volume fraction of MSP, we find that the instrument would collect of the order of $10^{14}$ to $10^{15}$ amu of refractory MSP particles. The estimate basis on the assumption that the ice components are melting and the flow conditions in the instruments are for typical atmospheric pressures at 85 km.

*Copyright statement.* TEXT



# 1 Introduction

The presence of dust particles influences physics and chemistry of the upper atmosphere in the meteor ablation zone and below. Meteors ablation is a source of dust particles in the upper atmosphere. The remnants of meteor ablation that prevail in the mesosphere condense to nm-sized particles, denoted as Meteoric Smoke Particles (MSP). Theory suggests that MSP act

as condensation nuclei for ice particles, which form during summer months around the mesopause at high and mid latitudes. The MSP and the ice particles are related to mesospheric phenomena such as the Noctilucent Clouds, Polar Mesospheric Summer and Winter Echoes (PMSE/PMWE)(see (Rapp and Lübken, 2004; Plane et al., 2015). Due to their altitude location, the only means of in situ measurement is with rocket experiments and the dust particles were detected using electric charge measurements in Faraday-cup instruments. There is little known about the MSP composition except for extinction estimates

based on satellite observations (Hervig et al. (2012)) and estimates of the material work function based on rocket observations (Rapp et al. (2012)). There have been several attempts to collect MSP with probes on rockets, but no conclusive results have been reported from their laboratory analysis so far. A detailed description of sample collection with the MAGIC instrument is given by Hedin et al. (2014). UiT have proposed a new sample collector, the MEteoric Smoke Sampler (MESS) (Havnes et al., 2015). The instrument is designed to collect large ice particles, in order to analyze the refractory MSP that they contain and

that remain in the sample collector after evaporation of the ice component. The expected advantage of this approach is that the collection of large ice particles is less influenced by the airflow conditions at the rocket payload than the smaller MSP.

In this paper we consider a possible design of MESS and investigate the conditions of dust entering the detector. After describing the basic design, we simulate the airflow for different atmospheric conditions. We study the motion of particles in the airflow in and around the detector to investigate the dust that directly reaches the collection area as well as the trajectories

of particles that reach the collection area after collision with the funnel. A model is introduced to estimate the fragmentation at the funnel. Based on these considerations we estimate the amount of dust that could be collected assuming dust densities obtained from observational models. We also find preferred rocket conditions for the dust collection.





## 2 Concept description

MESS is a rocket instrument, intended to collect dust particles in the mesosphere. Located on the top deck of the payload it is
exposed to the airflow caused by the rocket motion which will carry the particles into the instrument. The experiment idea, first
proposed in Havnes et al. (2015), is to collect large ice particles which are less influenced by the airflow around the payload
and increase the amount of collected material by building the instrument with a funnel. Figure 1 on the top shows a preliminary
sketch of the instrument and on the bottom the Mach number from simulations discussed below. The funnel opening has a
diameter of 5 cm, an angle of $75°$, and total height of 6 cm. The particles are collected in the bottom of the funnel, which has
a diameter of 1.8 cm and an area of $A_{coll}{=}254.5\ mm^2$. The collection area is kept small since this determines the size of the
closure device which is a critical component to be designed for sea recovery. The funnel increases the sampling area by a factor
of 7.7. The collection surface has not been decided yet, but we have looked into solutions with TEM grids, as was used in the
MAGIC instrument (Hedin et al., 2014). In the mesospheric altitudes the air density is still significant and rocket velocities are
in the order of $\sim 1000\ ms^{-1}$. This will result in a bow shock in front of the instrument. The deflection of particles is discussed
below.

## 3 Model description

The model assumption that we make to study the flow of dust in the detector are illustrated in Figure 2. To consider the motion
of the dust particles, we first investigate the motion of the background gas and its interaction with the instrument under various
atmospheric conditions to obtain typical values for mass, number density and velocity of the air flow. We apply the derived
values to calculate the dust trajectories in the flow considering primary particles that directly hit the collecting surface and
secondary particles that are produced through collision with the funnel. We assume that the dust particles with radii 1 to 9 nm
are MSP. The larger particles we assume to contain water ice and 3% volume fraction MSP.

### 3.1 DSMC of gas flow

The rocket will traverse the mesosphere/lower thermosphere (MLT) region with a flow regime that transitions from continuum
flow where the air motion is considered as a continuous fluid, to free molecular flow described with collisionless motion of the
individual molecules. The Knudsen number, $Kn$, describes the rarefication in the gas as the ratio of the mean free path $\lambda$ and
characteristic dimensions of the rocket, L, the coninuum flow is defined for $Kn < 0.1$ and free molecular flow for $Kn >> 10$.
Direct Simulation Monte Carlo (DSMC) is a common tool used to study the rarefied gas dynamics. We use the DS2V program
based on DSMC methods, developed by Bird and Brady (1994). Imposing an axial symmetry we introduce a two dimensional
description of the instrument. This is based on a macroscopic approach analysing the behaviour of several thousand molecules
simultaneously through a sampled area. By averaging over each sampled area, the macroscopic flow properties can be found.
The input parameters to the program are the background gas density, temperature and velocity. In addition, specification
regarding surface properties of the instrument can be made. The output from the DS2V program are discrete values, and by





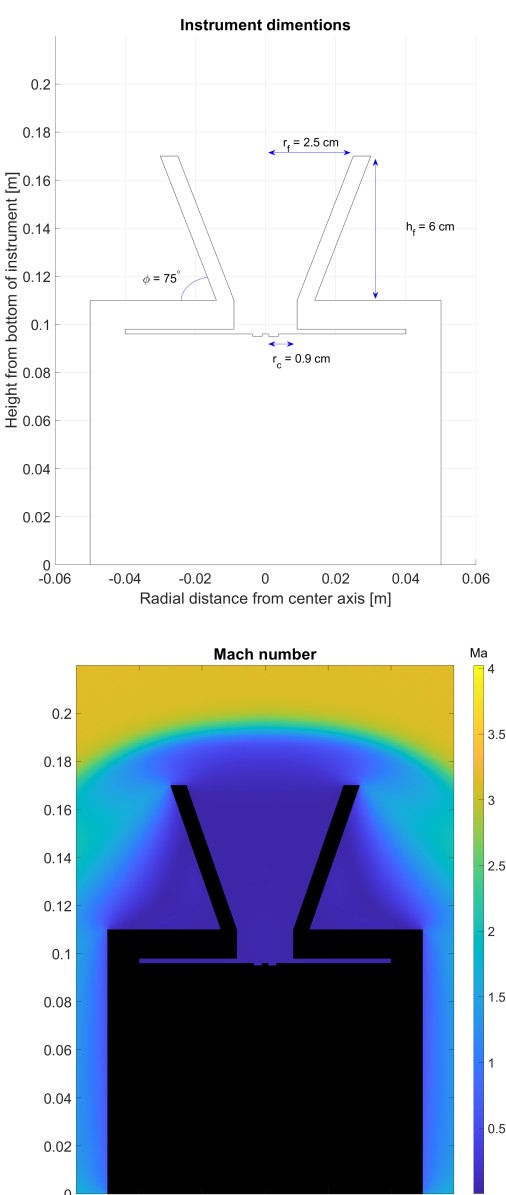

**Figure 1.** Top: Sketch of the MESS design assumed in this study. Bottom: Flow conditions for MESS derived from the DSMC for 85 km at $800 \ \mathrm{ms}^{-1}$, shown are Mach numbers in the airflow.

interpolating, we obtain a continuous function that can be used for the momentum equation. Figure 1 on the bottom show the

rarefication of the neutral gas at 85 km for a rocket velocity of $800 \ \mathrm{ms}^{-1}$.





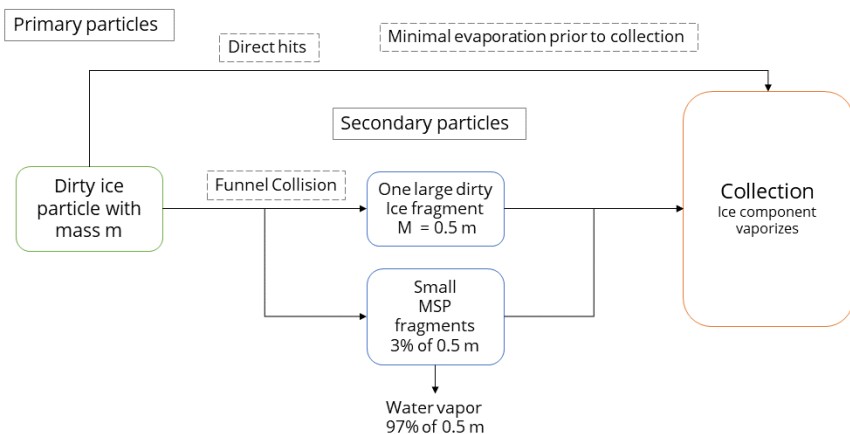

**Figure 2.** Flowchart of the model calculations.

## 3.2 Particle motion

Collisions with neutrals govern the motion of the dust particles. In the mesospheric region, the ionization is low, and we have neglected electric effects. There are several models used to introduce dust to the flow. We use the model presented in Antonsen and Havnes (2015), based on work by Horányi et al. (1999), Hedin et al. (2007) and Smirnov et al. (2007) which presents a drag

force on the dust which accounts for motion in "all" flow regimes. The rocket will be the reference frame of the simulations, and the initial velocity of the dust is the rocket velocity. Equation 1 states the drag force.

$$m_d \frac{d\boldsymbol{v_d}}{dt} = \chi \pi r_d^2 m_g n_g v_{th,g}(\boldsymbol{v_g} - \boldsymbol{v_d}) \frac{1}{u} \left[ \frac{1}{\sqrt{pi}} \left( u + \frac{1}{2u} \right) \exp(-u^2) + \left( 1 + u^2 - \frac{1}{4u^2} \right) \mathrm{erf}(u) \right] \tag{1}$$

Here $m_g$, $n_g$, $v_{th,g}$ and $\boldsymbol{v_g}$ are the mass, density, thermal and mean velocity of the background gas, where $m_g$ is assumed to

be constant, and $n_g$, $v_{th,g}$, $\boldsymbol{v_g}$ are determined by the DSMC. The dust parameters $r_d$ and $\boldsymbol{v_d}$ are radius and velocity respectively. The geometry factor given by $\chi$ is set to 1, assuming spherical particles. With the assumption of spherical particle, the dust mass is given as $\frac{4}{3}\pi r_d^3 \rho_d$, where $\rho_d$ is the dust density. The underlying assumptions are that the mass of the dust $m_d$ has to be much larger compared to the mass of the individual air molecules $m_g$, and a specular reflection of neutrals in collision with dust grains. We will only consider particles, both secondary and primary, of sizes above 0.8 nm.


**Table 1.** Parameters of dust and neutral atmosphere used in the calculations

| Parameter | Description | Value |
|---|---|---|
| $\chi$ | Shape factor | 1 |
| $r_d$ | Dust radius | 0.8-50 nm |
| $\boldsymbol{v_d}$ | Dust velocity | Output |
| $m_d$ | Dust mass | $\frac{4}{3}\pi r_d^3 \rho_d$, dependent on composition |
| $\rho_{d,ice}$ | Ice mass density | 1 g cm$^{-3}$ |
| $\rho_{d,msp}$ | MSP mass density | 3 g cm$^{-3}$ |
| $m_g$ | Mean molecular mass of gas | 29 amu |
| $n_g$ | Neutral gas number density | From DSMC |
| $v_{th,g}$ | Thermal velocity of gas | From DSMC |
| $\boldsymbol{v_g}$ | Mean gas velocity | From DSMC |
| u | Relative Mach number of dust | $|\boldsymbol{v_g} - \boldsymbol{v_d}|/v_{th,g}$ |

## 3.3 Funnel collision model

A large portion of the incoming primary particles will collide with the funnel wall. The interaction process is not well known, but we base our assumption on work by Tomsic et al. (2003), who have conducted several ice collision experiments and molecular simulations. They considered pure ice collisions with metal, for a range of impact velocities and temperatures. They found that for large incident angles with the surface normal, the particle has a high probability of one large fragment surviving the collision. Based on this we assume that for the ice particle collisions, one large fragment, with half the mass of the primary particle survives. The rest of the particle will be fragmented, following a fragmentation distribution based on the work by Antonsen et al. (2020), which states that the fragments will follow an $r^{-3}$ distribution, where r is the radius. We assume that the ice component of the fragments evaporates, and only the pure MSP are left. The fragments, or as we will call them, the secondary particles, are then traced by the method described in the previous section.

## 3.4 Mass Estimate

By using previous estimates on the number density of ice or MSP we can estimate the total mass of the samples. We will use a straightforward method, assuming a known number density, and estimating the sampled volume.

$$m_{collected} = A_{coll}\,\Delta h\,n_d\,V_{dust}\,\rho_{d,msp}\,\alpha\,\sigma \tag{2}$$

Where $A_{coll}, \Delta h$ and $V_{dust}$ are the collection area, sample altitude and volume of dust respectively. Volume given as $\frac{4}{3}\pi r_d^3$, assuming spherical particles. The dust number density is given as $n_d$, and $\rho_{d,msp}$ is the mass density of MSP. We denote the filling factor as $\alpha$ and the collection efficiency as $\sigma$.



**Table 2.** Input parameters for airflow calculations.

| Alt [km] | $n_g$ $[m^{-3}]$ | Temperature [K] | Initial velocity $[ms^{-1}]$ |
|----------|------------------|-----------------|------------------------------|
| 80 | $6.0154 \times 10^{20}$ | 169 | 800, 1000, 1200 |
| 85 | $2.2914 \times 10^{20}$ | 137 | 800, 1000, 1200 |

## 4 Results

In this section, we will present the results of the simulations and considerations. NLCs and PMSE occur at the altitudes 80-85 km, corresponding to altitudes where the conditions are such that ice can exist. To limit the amount of simulation we will only consider the altitudes, 80 and 85 km, representing the limits of our region of interest. We have considered the velocities 800, 1000 and 1200 $\mathrm{ms}^{-1}$.

### 4.1 Neutral gas flow

Input parameters for the neutral gas simulations are temperature, neutral air density and velocity (see table 2). Temperature and air density are taken from the MSIS-E-90 model (Chulaki, 2020). The rocket velocity varies in these altitudes and depends on the apogee of the rocket. For a mesospheric rocket with an apogee between 110-130 km we can expect a velocity around 1000 $\mathrm{ms}^{-1}$ through the region. We also investigate how rocket velocity variation will influence the dust trajectories.

At 80 km we have used a temperature of 169 K and a neutral air density of $6.0154 \times 10^{20} \mathrm{m}^{-3}$. Figure 3 shows the calculated neutral density distribution in the instrument for 800, 1000 and 1200 $\mathrm{ms}^{-1}$ from left to right respectively. The density is normalized with the maximum density reached inside the detector for all velocities, which is $9.2377 \times 10^{21}$ $\mathrm{m}^{-3}$. For 85 km the temperature was set to 137 K and a neutral air density of $2.2914 \times 10^{20}$ $\mathrm{m}^{-3}$, density decreases by a factor of $\sim 4$ over 5 km, and temperature decreases by $\sim 30$ K in comparison to the conditions at 80 km. Figure 4 shows the neutral density distribution in the instrument, obtained for 85 km conditions. The density is normalized with the maximum density reached found to be $3.5259 \times 10^{21}$ $\mathrm{m}^{-3}$. Note that the colorbars are different for the cases shown.

There is a significant increase in the neutral density inside the instrument by an order of magnitude, compared to the background. The simulations show that as the rocket velocity increases, the density inside the instrument also increases, and the maximum density for 800 $\mathrm{ms}^{-1}$ is only half of that for 1200 $\mathrm{ms}^{-1}$. This increase will cause an increase in the drag on the dust particles, which limits the size range of particles able to hit the collecting surface, as we will see in the following section. Another result seen from the simulations is that the air density increases from the instrument center to the funnel wall. Because of the lower background density at 85 km, we suggest to aim at collection around 85 km altitude and with a low rocket velocity. The simulations also show an increase in the air temperature, which may affect the ice particles. However, we will not go into detail on this at this stage.



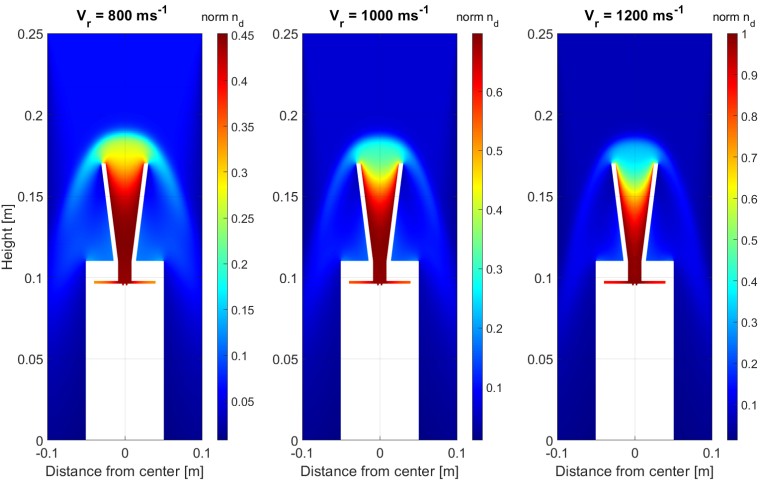

**Figure 3.** DSMC results for 80 km showing the calculated neutral density. Shown for velocities of $800\ \text{ms}^{-1}$, $1000\ \text{ms}^{-1}$ and $1200\ \text{ms}^{-1}$, left to right respectively. These values are normalized by $n_{\text{d,max}}{=}9.2377 \times 10^{21}\ \text{m}^{-3}$. Note that the color scales are different in the plots.

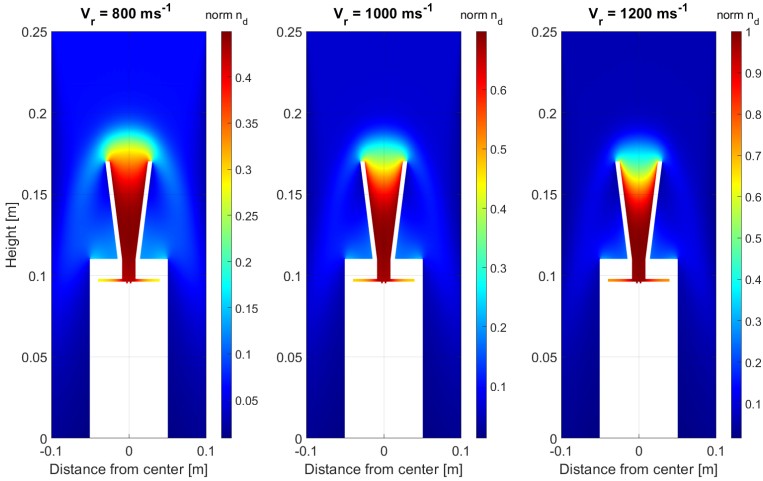

**Figure 4.** DSMC results for 85 km showing the neutral density. Shown for velocities of $800\ \text{ms}^{-1}$, $1000\ \text{ms}^{-1}$ and $1200\ \text{ms}^{-1}$, left to right respectively. These values are normalized by $n_{\text{d,max}}{=}3.5259 \times 10^{21}\ \text{m}^{-3}$. Note that the color scales are different in the plots.

## 4.2 Dust motion in instrument

The DS2V yield a significant decrease in density from 80 to 85 km, and an increase with rocket velocity from $800\ \text{ms}^{-1}$ to $1200\ \text{ms}^{-1}$. By using the results in the equation of motion, and solving it numerically using the 4th order Runge-Kutta method, we will see how the motion of the particles is affected. We distinguish between primary and secondary particles and a range






of sizes. We start by looking at the motion of the primary particles. These results where found tracing 25 primary particles and their fragments, for each size.

### 4.2.1 Primary particle trajectories

The primary particles enter the instrument without collision at the funnel. To study their trajectories, we assume small particles
with radii 1-9 nm and bulk density of $3\,\mathrm{g\,cm^{-3}}$ describing pure MSP and particles with radii 10 to 50 nm with bulk density $1\,\mathrm{g\,cm^{-3}}$ which contain water ice and MSP.

Figure 5 shows the trajectories at 80 km rocket height and for rocket velocities 800, 1000 and $1200\,\mathrm{ms^{-1}}$. Shown are the trajectories for particles with radii 1, 5 and 9 nm, the lightest color indicates the smallest, and darkest color the largest particles. We do not show the larger particles that are less affected by the airflow. The primary particles are those that enter the central
part of the detector within 9 mm from its symmetry axis. One can see that at rocket velocity $1200\,\mathrm{ms^{-1}}$ and $1000\,\mathrm{ms^{-1}}$ none of the trajectories shown here reach the collection area. This is the result of the increase in neutral density, seen in the previous section. The increase in neutral density causes an increase in drag on the particles. For the 800 m/s rocket velocity, a fraction of the trajectories shown reaches the collection area. The size cut-off for primary particles reaching the collection area that we find with the calculations is between 8-9 nm radius for $800\,\mathrm{ms^{-1}}$ rocket velocity and between 10 and 15 nm at $1200\,\mathrm{ms^{-1}}$
rocket velocity.

Figure 6 shows the trajectories at 85 km rocket height and for rocket velocities 800, 1000 and $1200\,\mathrm{ms^{-1}}$ and for dust sizes of 1, 3 and 9 nm. The air density in the instrument is lower at 85 km than at 80 km. Hence, the particles are less decelerated and more particles reach the collection area. In all three cases, the smallest particles that reach the collection are have radii between 3 and 5 nm. For the smaller particles, the detection efficiency depends on the distance from the instrument symmetry
axis at which the particles enter and the probability to reach the collection area is highest for particles that enter the instrument close to its central symmetry axis, since those particles are less decelerated. This is a result of the gas distribution seen in figure 3 and 4 which indicates that the drag on the particles varies with the distance from the instrument's symmetry axis.

With all the assumptions involved, we can conclude that in the considered range of velocities, assuming no angle of attack, the collection efficiency of primary particles with radii larger than roughly 10 nm is 1 in the suggested instrument and that it
thus is suitable to collect large ice particles. Calculations show that at 80 km, the smaller particles are more strongly affected by the background gas. Since we expect that the number density of MSP with radii larger 10 nm is very small, we can assume that for the conditions at 80 km rocket height, only few pure MSP will be collected with the MESS instrument and the dimensions considered. The simulations also suggest that at 85 km, pure MSP down to 3 nm are able to reach the collection surface, and 1 nm particles are not deflected at the entrance, but stopped inside the instrument.The calculations were made for dust particles
that enter the instrument parallel to the symmetry axis, while rocket payload typically are tilted to the velocity of the rocket and this angle of attack will reduce the flux of primary particles.





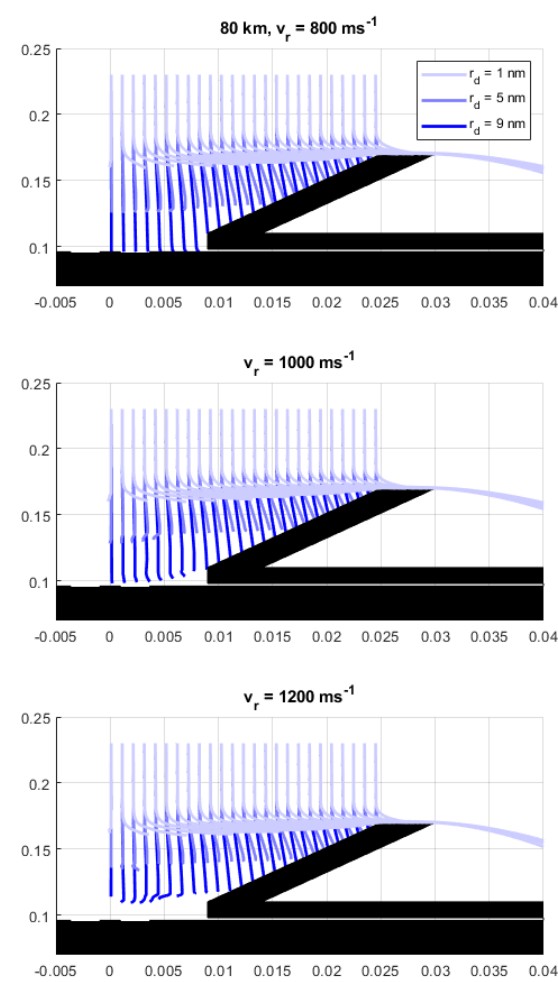

**Figure 5.** Primary particle trajectories for 80 km for rocket velocities of 800, 1000 and 1200 $\mathrm{ms}^{-1}$.

### 4.2.2 Secondary particle trajectory

We assume the impacting particles with radii smaller 9 nm are MSP and the larger particles are ice particles containing MSP. Secondary particles are produced through the collision of mesospheric dust with the funnel. As was shown above, a large

fraction of small particles is deflected in the instrument, while the larger particles are less affected by the airflow and hence will hit the funnel. Hervig et al. (2012) have reported a filling factor of maximum 3% of MSP. This corresponds to $\sim 270$ of 1 nm MSP in one 25 nm sized particle. Research by Tomsic et al. (2003) suggests a high probability of fragmentation when nm sized





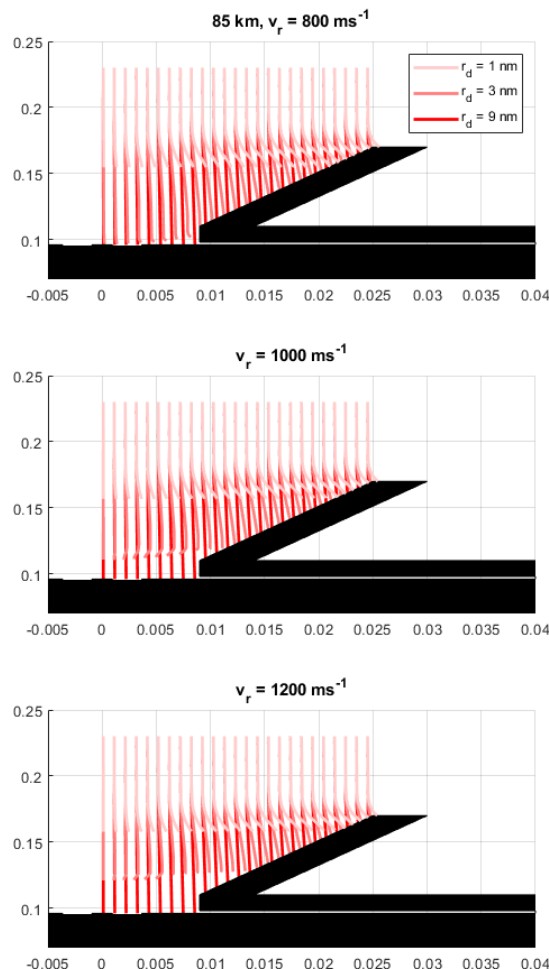

**Figure 6.** Primary particle trajectories for 85 km for rocket velocities of 800, 1000 and 1200 $\mathrm{ms}^{-1}$

particles collide with a metal surface. We make the simplifying assumption that half of the particle mass is contained in a large fragment containing ice and dust. The other half of the mass is distributed in smaller fragments for which the ice components evaporate and only refractory particles remain. For the refractory particles, we assume radii between 0.8 and 3 nm, which is a typical size for MSP. At the same time, the condition that the mass of the dust particle is large in comparison to mass of air particles is fulfilled. The scattering angle is randomized between 4 - 8 degrees from the surface, and we assume that 60% of the energy is conserved in the collision. The impact velocity of the primary particles upon colliding with the funnel wall is in the range $600-700\ \mathrm{ms}^{-1}$ for an initial velocity of $800\ \mathrm{ms}^{-1}$, and around $800\ \mathrm{ms}^{-1}$ for an initial velocity of $1200\ \mathrm{ms}^{-1}$






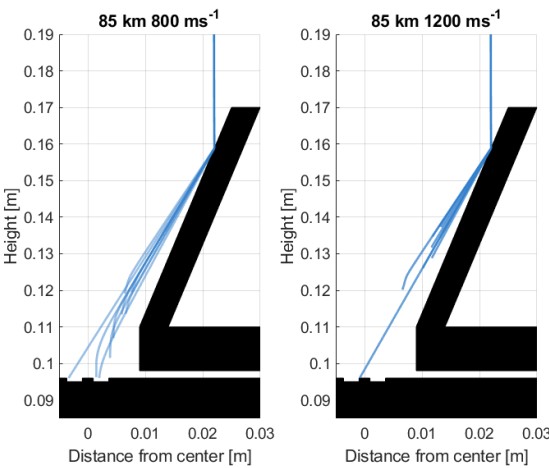

**Figure 7.** Secondary particle trajectories for a primary particle of 25 nm. Background conditions with altitude of 80 km and $1200\ \mathrm{ms}^{-1}$.

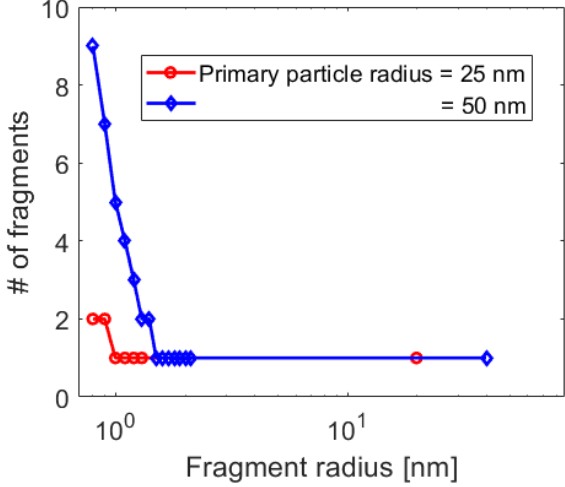

**Figure 8.** Fragment size distribution. Plotted for 25 and 50 nm in red and blue respectively.

Figure 7 shows the trajectories of primary particles of 25 nm colliding with the funnel wall and the resulting trajectories for the secondary particles. The trajectories are for an altitude of 85 km with rocket velocity of 800 and $1200\ \mathrm{ms}^{-1}$, from left to right respectively. Figure 8 shows the fragment distribution for a primary particle of 25 nm and 50 nm shown in red and blue respectively. By looking at the trajectories we again see the same effect on the smaller secondary particles as we saw for the smaller primary ones. For a higher velocity the particles are decelerated much faster than at lower velocity.

The simulations suggest that a large fraction of smaller fragments are stopped by the neutral air. The particles that are stopped will likely remain in the instrument, and could reach the collection area.





### 4.3 Estimate of collected mass

Based on previous observations we can estimate the collected amount of dust in the instrument, which is important for the choice of laboratory analysis. Here we focus on an altitude of 85 km, with an estimated number density of ice particles $n_d$=300 $m^{-3}$ for a mean particle radius of 25 nm (Kiliani et al., 2015), and a filling factor $\alpha$ of 3% for MSP contained in the particle.

First we consider the collection of primary particles of radius 25 nm. Simulations suggest a collection efficiency of 1 at 85 km altitude and for rocket velocities 800 $ms^{-1}$ . Collecting particles at a height interval of 1 km and collection area $A_{coll}$, the amount of 25 nm ice particles in the traversed volume is $\sim 7.6 \times 10^7$, corresponding to a mass of $\sim 2.7 \times 10^{14}$ amu. The trajectory simulations above suggest that 25 nm ice particles are large enough so that the drag in the airflow can be neglected. Heating of the particles may however lead to some mass loss.

Now we make an estimate for the secondary particles, which is much more uncertain, since it depends on the fragmentation process. The secondary particle, or annular sampling area is $A_f - A_{coll}$=17.1 $cm^2$. Using equation 2 and considering the same parameters as above, the estimated amount of material from secondary particle collection is $1.8 \times 10^{15}$ amu for the ideal case were all secondary particles are collected. This would increase the total amount of collected dust by almost one order of magnitude. The simulations however, suggest that a fraction of the smallest particles, especially those generated close to the opening, will not reach the collection area.

We estimate the total mass collection, by looking at a sampling layer in the range 0.5-4 km and collection efficiencies from 0.1 to 1. Here we should note that the sampling layer of the instrument may be much larger than 4 km, but the we assume that the dust densities outside this region is negligibly small. In the previous section we found that the collection efficiency is hard to determine, given all the variables. Our estimates suggest that it is around 0.5-0.8. Figure 9 shows three plots from left to right: the total collected mass, collected mass per collection area, and the number of particles per collection area, assuming an average size of 1 nm. The total collected mass is in the order of $10^{15}$ amu. If the collection efficiency is as low as 0.1, the expected mass would be $\sim 0.9 \times 10^{15}$ amu for an opening interval of 4 km. In terms of how much mass expected per $mm^2$ at the collection area, we can see from the middle figure that it is in the range of $10^{12} - 10^{13}$ amu per $mm^2$. Converting this to particles, again, assuming an average particle size of 1 nm, leads to about $10^4$ particles per $mm^2$.

### 4.4 Estimate of impact energy

If we wish to sample the particles on material such as carbon foil, the impact energy might be important considering possible perforation of the foil. Figure 9 shows the impact energies, top for larger energies and bottom for smaller. Note the logarithmic scale. Top figure shows the larger primary and secondary particles, in red triangles and blue circles respectively. The primary particles have energies 5 times higher than secondary particles of similar size. This is a result of the energy loss in the funnel collision. We also see that large particles have energies that range over several orders of magnitude.

The figure at the bottom shows the impact energies for the small secondary fragments. The size distribution is the same as in figure 6. Here we see that the impact velocities are much lower. The vertically aligned values show that there is a range of





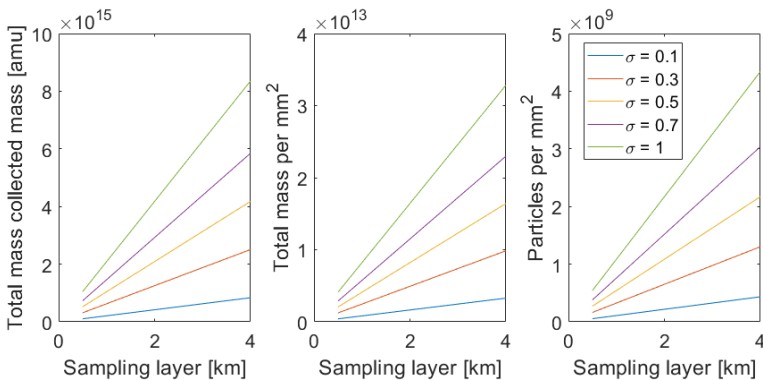

**Figure 9.** Mass as function of sampling distance. From left to right: Total collected mass, collected mass per collection area and particles per collection area. Values are given for different collection efficiencies $\sigma$.

impact energies for each size. The values depend on the initial fragment velocity after collision at the funnel, a parameter that has great uncertainty. The total range of impact energy estimated is from $10^{-3} - 10^{6}$ eV, as the impact mass is in such a large range.

## 5 Discussion and concluding remarks

The Proposed MESS design for collecting dust during rocket flight through a PMSE layer can return of the order of $10^{14}$ to $10^{15}$ amu of refractory MSP material assuming that the rocket samples a 0.5 - 4 km height interval of PMSE and that the collected ice particles contain a 3 percent volume fraction of refractory MSP. For particles that reach the collecting area, we find a range of impact energies between $10^{-3}$ eV and $10^{6}$ eV. This is partly a result of the range of particle sizes, but impact energies also vary considerably for particles of the same size. The collected particles are primary particles and secondary particles that were affected by fragmentation at the funnel, both lose their ice component either during the collection or later during the campaign.

Future work on a detailed design, could include studies of sticking efficiency and charging at the funnel, temperature studies and consideration of the collection surface material. The collision process at the funnel induces an uncertainty. The laboratory measurements that Tomsic et al. (2003) carried out with pure ice particles support our hypothesis that the dust particles will fragment at the funnel. We did not consider the cases that particles are reflected or sticking at the funnel surface, nor did we consider charge effects. Particles can carry an initial charge and can also be charged during fragmentation. The surface charge can affect the efficiency for sticking of particles to the funnel wall. Trajectories of the charged particles are also affected by electromagnetic forces. The rocket payloads tend to become charged in their trajectory through the ionosphere. However, in the mesosphere we assume that this charging will be small Lai (2011).

Our estimates were made for a normal impact direction of particles to the instrument. In the case that the flux direction is tilted relative to the normal (angle of attack), the impact angle from the surface normal is smaller for the particles impacting

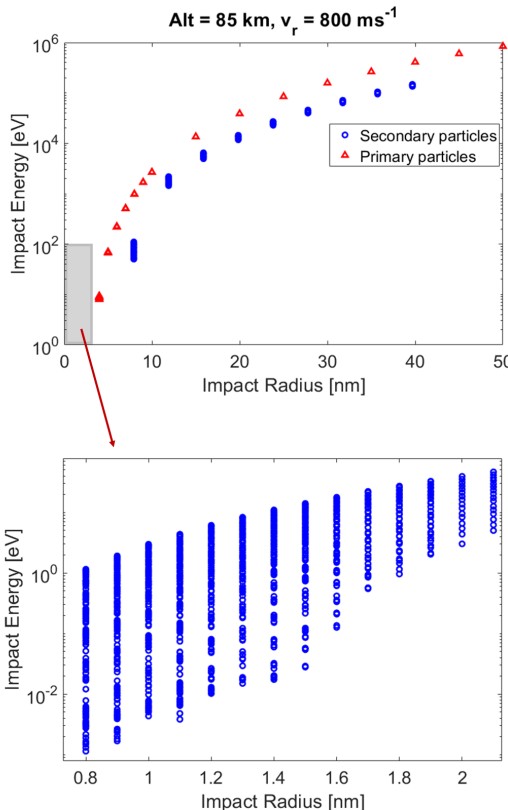

**Figure 10.** Impact energy for a range of primary and secondary particles

on the funnel. This has an influence on the fragmentation process. In the absence of a funnel, the direct flux of particles would
be reduced because of the rocket tilt relative to its direction of motion, which is less so the case with the funnel.

In summary, the discussed design of a sample collector combined with a funnel increases the amount of collected dust mass
by up to a factor of 7 because it has a larger sampling area. There is a cut-off for small particles that will not be collected. At
85 km, MESS will collect particles larger than roughly 4 nm radius. The cut-off for small particles is lower in the absence of a
funnel, but the sampling area would be reduced. Dust collection with MESS should aim toward the higher altitude of PMSE.
Our simulations suggest that for the same amount of dust in the atmosphere, a significantly higher amount of particles reach
the collecting area at an altitude of 85 km in comparison to 80 km. With increasing rocket velocity, the amount of background
gas in the instrument increases and so does the deceleration of particles in the instrument.

*Data availability.* The DS2V program can be found at http://www.gab.com.au/page4.html





*Acknowledgements.* This work was supported by the Research Council of Norway through grant numbers NFR 275503 and NFR 240065.

The publication charges for this article have been funded by a grant from the publication fund of UiT The Arctic University of Norway.



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
