# Peer review of "Investigating the Dust Flux in the Meteoric Smoke Sampler (MESS) Instrument for Sampling Dust in the Mesosphere"

_Atmospheric Measurement Techniques, 2020_

## Referee Comment (RC1) · Anonymous Referee #1 · 25 Aug 2020

Referee report on the paper amt-2020-278

"Investigating the Dust Flux in the Meteoric Smoke Sampler (MESS) Instrument for Sampling Dust in the Mesosphere"
by Henriette Trollvik, Ingrid Mann, Sveinung Olsen, and Yngve Eilertsen

submitted for publication in AMT

The paper aims at presenting a novel rocket-borne instrument for sampling of Meteoric Smoke Particles (MSP) in mesosphere and justifying its aerodynamic properties. The authors present some results of aerodynamic simulations for neutral gas surrounding the instrument flying at supersonic velocities and for MSP-flow through this environment into the instrument.

The paper is well structured and the results are clearly present. Nevertheless, there are some points which can be addressed in the frame of this study. Therefore, I recommend minor revision.

Herewith I suggest some possible improvements to be considered for the revised version of the manuscript. Since Copernicus publisher will anyway make language corrections, I will ignore typos and some weird formulations.

1. Results of DSMC simulations of the environment (Fig. 1 or 3 & 4) may additionally include flow streamlines to make it more clear for the reader to understand the deflection of MSPs by the surrounding gas flow.

2. The results demonstrated by Fig. 3 & 4 will be easier to understand if colorbars will be on the same scale (one color scale for all these plots).

3. Taking neutral temperature and density from MSISE-90 model as input for simulation is acceptable, however natural variability of these parameters must be taken into account. Thus, for instance, rocket-borne measurements at high northern latitudes [e.g., *Lübken et al.*, 1999; *Strelnikov et al.*, 2013] as well as e.g., lidar measurements in Antarctica [*Lübken et al.*, 2015] show temperature variability of $\sim$40-50 K at altitudes of interest for the manuscript. Also the neutral density variability as can be seen in Fig. 9 of *Strelnikov et al.* [2013], makes $4 \cdot 10^{20}$ to $7 \cdot 10^{20} \, \mathrm{m}^{-3}$ and approximately $1.5 \cdot 10^{20}$ to $3 \cdot 10^{20} \, \mathrm{m}^{-3}$ at 80 and 85 km altitude, respectively. This variability will contribute to uncertainties of the derived results (which is not addressed in the manuscript).

4. Simulation of MSP flow (Sec. 3.2) must be described in more details. For instance, that you (probably) start particle tracking (solving Eq.1) outside the shock front and stop if some conditions are met. This should further clarify, e.g., what happens to MSPs which do not hit the collecting surface (which I have not understood after reading the entire manuscript). You could also specify the grid used for these calculations, etc.

5. Sec. 3.4 (Mass Estimate) contains two parameters: filling factor $\alpha$ and the collection efficiency $\sigma$, which must be explained.

6. MSP trajectories in Fig. 5 & 6 are difficult to see (may be use of different colors can improve).

7. The manuscript makes an impression that only a single run of MSP flow simulation was made. This makes the results not quite reliable. Since these simulations have a probabilistic character, a certain number of trajectories must be calculated to gather an appropriate statistics. Thus, e.g., for assessment of MSP collection efficiency *Asmus et al.* [2017] simulated trajectories for 4000 particles.

8. The same is also true for the fragmentation study shown in Fig. 8; i.e. statistics and uncertainties are not shown.

9. The sentence in P.12 L.180: *"The particles that are stopped will likely remain in the instrument, and could reach the collection area."* need some explanation. Why and how it happens? Why not blown away during payload precession?

10. In Sec. 4.3 (Estimate of collected mass) authors refer to a model study by *Kiliani et al.* [2015] in context of justification for their choice of MSP parameters, which is not appropriate. Please, refer to original measurements.

11. Also, in many places of the manuscript references are missing. E.g., P.7 L.104 (existence of NLC/PMSE conditions), P.9 L.135-136 (for MSP densities), P.11 L.171 (typical MSP size), and similar statements where a certain value (so-called typical) is assigned to some parameter.

12. P.13 L.189 *"traversed volume is..."* units are missing.

13. p.13 L.193 I do not understand the statement *"The secondary particle, or annular sampling area is $A_f$"* and $A_f$ is not defined.

14. Also in this discussion (Sec. 4.3) uncertainties are not addressed.

15. *"Heating of the particles"* (e.g. P.13, L.191) is often mentioned in the manuscript, but never explained: why and how much (how fast) to expect.

16. P.13 L.209 must be Fig. 10

17. Discussion may address many uncertainties. E.g., angle of attack, which is somehow mentioned in the manuscript, but not quantified. Such values would help to define flight parameters needed for a judgment whether the instrument is suitable for a particular mission. For example, what is the critical angle of attack, what are velocity limits (rocket apogee) for presumably satisfactory MSP collection in the given altitude region. How the results are sensitive to sizes of ice particles? Will any PMSE be enough for a successful MSP sampling or bigger particles (NLC) are needed.

18. The conclusion inferred from simulations and often mentioned in the manuscript, that MSP collection is more efficient at 85 km compared to 80 km is already long time a well known result [e.g., *Horanyi et al.*, 1999; *Hedin et al.*, 2007; *Strelnikova et al.*, 2009; *Asmus et al.*, 2017].

19. Abbreviations MAGIC (instrument) and TEM (grids) in the beginning of the manuscript are not described.

**References**

Asmus, H., T. Staszak, B. Strelnikov, F.-J. Lübken, M. Friedrich, and M. Rapp, Estimate of size distribution of charged MSPs measured in situ in winter during the WADIS-2 sounding rocket campaign, *Ann. Geophys.*, pp. 979–998, doi:10.5194/angeo-35-979-2017, 2017.

Hedin, J., J. Gumbel, and M. Rapp, On the efficiency of rocket-borne particle detection in the mesosphere, *Atmospheric Chemistry & Physics*, *7*, 3701–3711, doi:10.5194/acpd-7-1183-2007, 2007.

Horanyi, M., J. Gumbel, G. Witt, and S. Robertson, Simulation of rocket-borne particle measurements in the mesosphere, *Geophysical Research Letters - GEOPHYS RES LETT*, *26*, 1537–1540, doi:10.1029/1999GL900298, 1999.

Kiliani, J., G. Baumgarten, F.-J. Lübken, and U. Berger, Impact of particle shape on the morphology of noctilucent clouds, *Atmos. Chem. Phys.,*, *15*, 12,897–12,907, doi:10.5194/acp-15-12897-2015, 2015.

Lübken, F.-J., M. Rapp, J. Siebert, and K. H. Fricke, The thermal and dynamical state of the upper atmosphere during the first flight of the NLTE campaign, in *Proceedings of the 14th ESA Symposium on European Rocket and Balloon Programmes and Related Research*, vol. ESA SP–437, edited by B. Kaldeich-Schürmann, pp. 363–368, Potsdam, Germany, 1999.

Lübken, F.-J., J. Höffner, T. P. Viehl, E. Becker, R. Latteck, B. Kaifler, D. Murphy, and R. J. Morris, Winter/summer transition in the Antarctic polar mesopause region, *J. Geophys. Res.*, *120*, 12, 394–12,409, doi:10.1002/2015JD023928, 2015.

Strelnikov, B., M. Rapp, and F.-J. Lübken, In-situ density measurements in the mesosphere/lower thermosphere region with the TOTAL and CONE instruments, in *An Introduction to Space Instrumentation*, pp. 1–11, TERRAPUB, doi:10.5047/aisi.001, 2013.

Strelnikova, I., et al., Measurements of meteor smoke particles during the ECOMA-2006 campaign: 2. Results, *Journal of Atmospheric and Solar-Terrestrial Physics*, *71*(3-4), 486–496, 2009.

---

## Referee Comment (RC2) · Anonymous Referee #2 · 2 Sep 2020

The paper investigates the movement of particles in a conceptual instrument for collecting dust particles during a sounding rocket flight and later sea retrieval. This is accomplished by combining several models and simulations. For a future deployment, it is important to understand how dust moves from the atmosphere into the instrument and onto the collecting surface.

General comments:

At the present stage of development, the investigation is clearly aiming to find the boundaries of the design, as no closing mechanism or collecting surface is defined. It would be nice to elaborate this more clearly, e.g. which collection principles exists and

what are their requirements. What would be other requirements or degrees freedom, e.g. from the rocket and environment?

Further I would recommend focusing less on the work that has clearly been done, but more on the meaning of it. For example in Fig. 2, the shock sure looks nice but what is it that you want to clarify to the reader, especially as the Mach number is not further discussed? Or the different trajectories in Fig. 5 & 6, what should the reader see in those figures? Even if crude, it would be more helpful to give e.g. the collection efficiency, in a table or figure, to see more clearly which altitude and velocity is preferred.

Why are only 80 and 85 km simulated? The reader is forced through half the paper before knowing in the results section that PMSE are limited to those altitudes. In Rapp and Lübken (2004) the altitude range is given with 80 to 90 km for PMSE with a clear peak at 85 km, while NLCs (large ice particles) peaking between 80 and 85 km. Thus particle size is a function of altitude, with the heaviest being lowest. This was not considered in the present paper and it feels like 90 km is missing in the simulations, especially if one could assume different particle sizes at different altitudes, which would also lead to different ratios of primary and secondary particles.

In 4.2.1 it is stated that primary particles (not colliding with funnel) are simulated, but in Figures 5 & 6 plenty of particles hit the funnel? It is further not clear, from the figures if they reach the surface or if they just move out of the plane? As the pressure regime is within the Knudsen flow (if i am not mistaken), particle trajectories should have more of a statistical outcome? 8 or 9 primary particle trajectories could be not enough?

In the results section a lot of work seems to be swept away by assuming a collection efficiency of unity and calculate the total amount of particles when flying a known collection area through a layer of an assumed density and then vary layer thickness and collection efficiency without taking into account the simulation results or other constraints.

Usually the assumption of an angle of attack = 0° is always wrong, as most rockets do not have attitude control. Maybe this could be more reasoned as insignificant for typical angles of attack in the given scenarios. A slower rocket at higher altitude as proposed might show significantly higher angles of attack.

If best results are obtained for lower pressures, could there be a more optimised shape of the funnel?

Line comments:

Line 11 citations are usually avoided in the abstract

Line 18 Meteor ablation

Line 22 which altitudes specifically

Line 23-24 split sentence, reference for the Faraday cup measurements?

Line 37 rocket conditions sounds odd, measurement conditions or something

Line 56 reference ?

Line 63 Knudsen number introduced not further mentioned in text

Line 92 radius of what

Line 103 PMSE altitudes should be in the introduction, may be reason the rocket velocities

Line 128 what density? more specific

Line 137 & 146 & 157 & elsewhere "rocket height", maybe a bit misleading: "altitude" could be more appropriate.

Line 148 area?

Line 150 maybe split the sentence

Line 185 cm-3

Line 199 the

Line 205 3 different units are given, maybe describe which one would be a good criterion and why.

Line 208 In introduction it was a TEM grid, now a carbon foil, maybe that can be better introduced

Line 209 Figure 10 not Figure 9

Line 232 formatting of citation

Line 238 why not make it larger for even more particles? why is it a reasonable funnel size or aspect ratio (diameter / funnel length)? Why not sample as much as possible? e.g. 80 to 90 km

Line 243 the energies of the particles increase with the square of velocity, why is the number density the dominating factor and why does this not just increase or decrease a probability, e.g. the collection efficiency via number of air molecule collisions?

Figure comments: Figure 1,3,4,5,6,7: could at least one (preferably more) figure use the same scale on x and y ? Each figure group has a different scaling.

Figure 1: colour map or scale not suitable as e.g. Ma=1 is hardly visible. The Mach number is not discussed in the text. Why 800 m/s and 85 km? Half the figure is white or black. The lower panel has no axis labels, typo in upper panel.

Figure 3 & 4: Colour scales should be comparable, maybe normalize to ambient density as this is constant between the panels. consider a log colour scale.

Figure 5 & 6: Labels missing! Chosen colours make it difficult to distinguish between sizes. 5 nm at 80 km and 3 nm at 85 km, why the change?

Figure 7: plot says 85, caption 80

Figure 10: Maybe plot x logarithmic too and combine both figures? What is meant with impact radius? The lower panel seems to have a distribution, consider plotting the extremes or the mean.

**[AMTD](https://www.atmospheric-measurement-techniques.net/)**

---

## Author Comment (AC1) · 4 Nov 2020

**Response to Anonymous Referee #1**

We thank the reviewer for the constructive comments and will modify the manuscript accordingly. Our response to the review is given below inserted after each comment. The reviewer's comments are in *cursive* and our response in regular letters.

*The paper aims at presenting a novel rocket-borne instrument for sampling of Meteoric Smoke Particles (MSP) in mesosphere and justifying its aerodynamic properties. The authors present some results of aerodynamic simulations for neutral gas surrounding the instrument ying at supersonic velocities and for MSP-ow through this environment into the instrument. The paper is well structured and the results are clearly present. Nevertheless, there are some points which can be addressed in the frame of this study. Therefore, I recommend minor revision. Herewith I suggest some possible improvements to be considered for the revised version of the manuscript. Since Copernicus publisher will anyway make language corrections, I will ignore typos and some weird formulations.*

*1. Results of DSMC simulations of the environment (Fig. 1 or 3 & 4) may additionally include ow streamlines to make it more clear for the reader to understand the deflection of MSPs by the surrounding gas flow.*

- The figures will be changed accordingly including streamlines.

*2. The results demonstrated by Fig. 3 & 4 will be easier to understand if colorbars will be on the same scale (one color scale for all these plots).*

- The figures will be changed accordingly.

*3. Taking neutral temperature and density from MSISE-90 model as input for simulation is acceptable, however natural variability of these parameters must be taken into account. Thus, for instance, rocket-borne measurements at high northern latitudes [e.g., Lübken et al., 1999; Strelnikov et al., 2013] as well as e.g., lidar measurements in Antarctica [Lübken et al., 2015] show temperature variability of ~40-50K at altitudes of interest for the manuscript. Also the neutral density variability as can be seen in Fig. 9 of Strelnikov et al. [2013], makes $4 \cdot 10^{20}$ to $7 \cdot 10^{20}$ $m^{-3}$ and approximately $1.5 \cdot 10^{20}$ to $3 \cdot 10^{20}$ $m^{-3}$ at 80 and 85km altitude, respectively. This variability will contribute to uncertainties of the derived results (which is not addressed in the manuscript).*

- Since the purpose of our investigation is to understand the conditions for dust collections in general, we are interested in finding a possible range of results, rather than the result for a certain atmospheric condition. We will point this out in the paper, and we will also discuss the atmospheric variability that – as the reviewer points out - will determine the uncertainty of our prediction.

*4. Simulation of MSP flow (Sec. 3.2) must be described in more details. For instance, that you (probably) start particle tracking (solving Eq.1) outside the shock front and stop if some*

*conditions are met. This should further clarify, e.g., what happens to MSPs which do not hit the collecting surface (which I have not understood after reading the entire manuscript). You could also specify the grid used for these calculations, etc.*

- We will describe the calculations in more detail including the stopping conditions. We will also elaborate on what happens to the particles which do not hit the collecting surface.

*5. Sec. 3.4 (Mass Estimate) contains two parameters: filling factor α and the collection efficiency σ, which must be explained.*

- Explanations will be added in text

*6. MSP trajectories in Fig. 5 & 6 are difficult to see (may be use of different colors can improve).*

- Figure changes have been made

*7. The manuscript makes an impression that only a single run of MSP flow simulation was made. This makes the results not quite reliable. Since these simulations have a probabilistic character, a certain number of trajectories must be calculated to gather an appropriate statistics. Thus, e.g., for assessment of MSP collection efficiency Asmus et al. [2017] simulated trajectories for 4000 particles.*

- In contrast to the work by Asmus et al., we do not aim to derive a number density or flux from observed event rates, but we aim to obtain a reasonable estimate of the dust mass and dust sizes of particles that would be collected with MESS, hence 1% accuracy given in the above-mentioned paper is not the goal here. The largest uncertainties of the estimate that we make lies in the uncertainty of predicting the atmospheric parameters for the flight. We now point this out in the text.
- In addition, our main results refer to large primary particles for which other studies showed that they are not affected by Brownian motion and therefore we do not deem it necessary to use more trajectories for the primary particle estimates. The descriptions of the smaller particles shown are meant to serve more as an illustration of the flight and instrument condition. Phrasing in the text will be changed to point this out and we show more results on the smaller fragment trajectories.

*8. The same is also true for the fragmentation study shown in Fig. 8; i.e. statistics and uncertainties are not shown.*

- The uncertainties of the estimate that we make lie uncertainty of predicting the atmospheric parameters for the flight. We now point this out in the text.

*9. The sentence in P.12 L.180: "The particles that are stopped will likely remain in the instrument, and could reach the collection area." Need some explanation. Why and how it happens? Why not blown away during payload precession?*

- We expect that some particles could leave the detector, we did not see any cases in our calculation and given the conditions, we expect it is more likely that they remain in the instrument. We will mention this in the text.

*10. In Sec. 4.3 (Estimate of collected mass) authors refer to a model study by Kiliani et al. [2015] in context of justification for their choice of MSP parameters, which is not appropriate. Please, refer to original measurements.*

- Since we want to estimate conditions for the entry into the instrument, it does not seem plausible to us to use in such an estimate results from in-situ measurements that could be biased by similar effects. We will however consider MSP parameters obtained from model calculations in addition to estimates from NLC observations.

*11. Also, in many places of the manuscript references are missing. E.g., P.7 L.104 (existence of NLC/PMSE conditions), P.9 L.135-136 (for MSP densities), P.11 L.171 (typical MSP size), and similar statements where a certain value (so-called typical) is assigned to some parameter.*

- We will check and correct the references.

*12. P.13 L.189 "traversed volume is..." units are missing.*

- We will correct this.

*13. p.13 L.193 I do not understand the statement "The secondary particle, or annular sampling area is $A_f$" and $A_f$ is not defined.*

- We will correct this.

*14. Also in this discussion (Sec. 4.3) uncertainties are not addressed.*

- We will discuss the uncertainties as outlined above.

*15. "Heating of the particles" (e.g. P.13, L.191) is often mentioned in the manuscript, but never explained: why and how much (how fast) to expect.*

- We will include calculations of the dust temperatures for typical cases and discuss the implications for the collection experiment.

*16. P.13 L.209 must be Fig. 10*

- We will revise this.

*17. Discussion may address many uncertainties. E.g., angle of attack, which is somehow mentioned in the manuscript, but not quantified. Such values would help to define flight parameters needed for a judgment whether the instrument is suitable for a particular mission. For example, what is the critical angle of attack, what are velocity limits (rocket apogee) for presumably satisfactory MSP collection in the given altitude region. How the results are*

*sensitive to sizes of ice particles? Will any PMSE be enough for a successful MSP sampling or bigger particles (NLC) are needed.*

- We will expand the discussion part as suggested.

*18. The conclusion inferred from simulations and often mentioned in the manuscript, that MSP collection is more efficient at 85km compared to 80km is already long time a well known result [e.g., Horanyi et al., 1999; Hedin et al., 2007; Strelnikova et al., 2009; Asmus et al., 2017].*

- We agree this was found before for other instruments and now clarify that in the discussion.

*19. Abbreviations MAGIC (instrument) and TEM (grids) in the beginning of the manuscript are not described.*

- We will check and explain abbreviations used.

**References:**

Asmus, H., T. Staszak, B. Strelnikov, F.-J. Lubken, M. Friedrich, and M. Rapp, Estimate of size distribution of charged MSPs measured in situ in winter during the WADIS-2 sounding rocket campaign, *Ann. Geophys.*, pp. 979-998, doi:10.5194/angeo-35-979-2017, 2017.

Hedin, J., J. Gumbel, and M. Rapp, On the efficiency of rocket-borne particle detection in the mesosphere, *Atmospheric Chemistry & Physics*, 7, 3701-3711, doi:10.5194/acpd-7-1183-2007, 2007.

Horanyi, M., J. Gumbel, G. Witt, and S. Robertson, Simulation of rocketborne particle measurements in the mesosphere, *Geophysical Research Letters - GEOPHYS RES LETT*, 26, 1537-1540, doi:10.1029/1999GL900298, 1999.

Kiliani, J., G. Baumgarten, F.-J. Lübken, and U. Berger, Impact of particle shape on the morphology of noctilucent clouds, *Atmos. Chem. Phys.*,, 15,12,897-12,907, doi:10.5194/acp-15-12897-2015, 2015.

Lübken, F.-J., M. Rapp, J. Siebert, and K. H. Fricke, The thermal and dynamical state of the upper atmosphere during the first flight of the NLTE campaign, in *Proceedings of the 14th ESA Symposium on European Rocket and Balloon Programmes and Related Research,* vol. ESA SP-437, edited by B. Kaldeich-Schürmann, pp. 363-368, Potsdam, Germany, 1999.

Lübken, F.-J., J. H• o_ner, T. P. Viehl, E. Becker, R. Latteck, B. Kaier, D. Murphy, and R. J. Morris, Winter/summer transition in the Antarctic polar mesopause region, *J. Geophys. Res.*, 120, 12, 394-12,409, doi: 10.1002/2015JD023928, 2015.

Strelnikov, B., M. Rapp, and F.-J. L• ubken, In-situ density measurements in the mesosphere/lower thermosphere region with the TOTAL and CONE instruments, in *An Introduction to Space Instrumentation*, pp. 1-11, TERRAPUB, doi:10.5047/aisi.001, 2013.

Strelnikova, I., et al., Measurements of meteor smoke particles during the ECOMA-2006 campaign: 2. Results, *Journal of Atmospheric and Solar-Terrestrial Physics*, 71 (3-4), 486-496, 2009.

---

## Author Comment (AC2) · 4 Nov 2020

**Response to Anonymous Referee #2**

We thank the reviewer for the constructive review which helps us to improve the manuscript. Our response is given below together with the reviewer comments. The reviewer comments are in *cursive* letters and our response in regular letters.

*The paper investigates the movement of particles in a conceptual instrument for collecting dust particles during a sounding rocket flight and later sea retrieval. This is accomplished by combining several models and simulations. For a future deployment, it is important to understand how dust moves from the atmosphere into the instrument and onto the collecting surface.*

*General comments:*
*At the present stage of development, the investigation is clearly aiming to find the boundaries of the design, as no closing mechanism or collecting surface is defined. It would be nice to elaborate this more clearly, e.g. which collection principles exists and what are their requirements. What would be other requirements or degrees freedom, e.g. from the rocket and environment?*

- We will expand the discussion part of the paper and we will refer to previous works of other groups that made sample collection experiments in order to justify our approach.

*Further I would recommend focusing less on the work that has clearly been done, but more on the meaning of it. For example in Fig. 2, the shock sure looks nice but what is it that you want to clarify to the reader, especially as the Mach number is not further discussed? Or the different trajectories in Fig. 5 & 6, what should the reader see in those figures? Even if crude, it would be more helpful to give e.g. the collection efficiency, in a table or figure, to see more clearly which altitude and velocity is preferred.*

- We will revise the manuscript following this comment and will discuss the presented figures in more detail.

*Why are only 80 and 85 km simulated? The reader is forced through half the paper before knowing in the results section that PMSE are limited to those altitudes. In Rapp and Lübken (2004) the altitude range is given with 80 to 90 km for PMSE with a clear peak at 85 km, while NLCs (large ice particles) peaking between 80 and 85 km. Thus particle size is a function of altitude, with the heaviest being lowest. This was not considered in the present paper and it feels like 90 km is missing in the simulations, especially if one could assume different particle sizes at different altitudes, which would also lead to different ratios of primary and secondary particles.*

- The parameters at 80 and 85 km were chosen as the boundaries within which noctilucent clouds are observed and hence large particles exist. We focus on this lower altitude because we expect larger particles that are present there are less influenced by the airflow and more easily collected. The funnel concept is chosen so that the collection area increases and at the same time the ice particles fragment when colliding with the funnel so that their fragments can be collected. We now state this in the introductory sections.

*In 4.2.1 it is stated that primary particles (not colliding with funnel) are simulated, but in Figures 5 & 6 plenty of particles hit the funnel? It is further not clear, from the figures if they reach the surface or if they just move out of the plane? As the pressure regime is within the Knudsen flow (if I am not*

*mistaken), particle trajectories should have more of a statistical outcome? 8 or 9 primary particle trajectories could be not enough?*

- We now address the statistical nature of the calculations. We also include a calculation using more primary trajectories for comparison. In a project paper recently made at UiT, it was found that the Brownian motion influences particle smaller 2 nm, but less so the larger once. From this we expect that the primary particles are sufficiently well described with the presented trajectory calculations for the estimate that we present which includes the large uncertainty of atmospheric conditions. We will clarify in the text how we define the primary and the secondary particles that we discuss.

*In the results section a lot of work seems to be swept away by assuming a collection efficiency of unity and calculate the total amount of particles when flying a known collection area through a layer of an assumed density and then vary layer thickness and collection efficiency without taking into account the simulation results or other constraints.*

- We will include estimates of the expected collected mass based on the calculation in combination with different assumed dust densities in the atmosphere. We will also discuss the results of the calculations in more detail

*Usually the assumption of an angle of attack = $0^o$ is always wrong, as most rockets do not have attitude control. Maybe this could be more reasoned as insignificant for typical angles of attack in the given scenarios. A slower rocket at higher altitude as proposed might show significantly higher angles of attack.*

- We also speculated about angles of attack for different flight conditions but were unable to get a clear statement from the experts on this. Since we have no clear information on the angle of attack it is hard to say whether it will be significant or not. We will point this uncertainty out in the discussion.

*If the best results are obtained for lower pressures, could there be a more optimized shape of the funnel?*

- The conical shape of the funnel was chosen because it allows us to collect fragments from the entire funnel area. Its dimensions are limited by other dust detectors in the strawman payload. We now mention this in the discussion.

We will thoroughly revise the manuscript considering all the line and figure comments.

*Line comments:*
*Line 11 citations are usually avoided in the abstract*
*Line 18 Meteor ablation*
*Line 22 which altitudes specifically*
*Line 23-24 split sentence, reference for the Faraday cup measurements?*
*Line 37 rocket conditions sounds odd, measurement conditions or something*
*Line 56 reference ?*
*Line 63 Knudsen number introduced not further mentioned in text*

*Line 92 radius of what*

*Line 103 PMSE altitudes should be in the introduction, may be reason the rocket velocities*

*Line 128 what density? more specific*

*Line 137 & 146 & 157 & elsewhere "rocket height", maybe a bit misleading: "altitude" could be more appropriate.*

*Line 148 area?*

*Line 150 maybe split the sentence*

*Line 185 cm-3*

*Line 199 the*

*Line 205 3 different units are given, maybe describe which one would be a good criterion and why.*

*Line 208 In introduction it was a TEM grid, now a carbon foil, maybe that can be better introduced*

*Line 209 Figure 10 not Figure 9*

*Line 232 formatting of citation*

*Line 238 why not make it larger for even more particles? why is it a reasonable funnel size or aspect ratio (diameter / funnel length)? Why not sample as much as possible? E.g. 80 to 90 km*

*Line 243 the energies of the particles increase with the square of velocity, why is the number density the dominating factor and why does this not just increase or decrease a probability, e.g. the collection efficiency via number of air molecule collisions?*

*Figure comments: Figure 1,3,4,5,6,7: could at least one (preferably more) figure use the same scale on x and y? Each figure group has a different scaling.*

*Figure 1: colour map or scale not suitable as e.g. Ma=1 is hardly visible. The Mach number is not discussed in the text. Why 800 m/s and 85 km? Half the figure is white or black. The lower panel has no axis labels, typo in upper panel.*

*Figure 3&4: Color scales should be comparable, maybe normalize to amient density as this is constant between the panels. Consider a log color scale.*

*Figure 5 & 6: Labels missing! Chosen colours make it difficult to distinguish between sizes. 5 nm at 80 km and 3 nm at 85 km, why the change?*

*Figure 7: plot says 85, caption 80*

*Figure 10: Maybe plot x logarithmic too and combine both figures? What is meant with impact radius? The lower panel seems to have a distribution, consider plotting the extremes or mean.*